# The Intrinsic Cardiac Nervous System: From Pathophysiology to Therapeutic Implications

**DOI:** 10.3390/biology13020105

**Published:** 2024-02-07

**Authors:** Giuseppe Giannino, Valentina Braia, Carola Griffith Brookles, Federico Giacobbe, Fabrizio D’Ascenzo, Filippo Angelini, Andrea Saglietto, Gaetano Maria De Ferrari, Veronica Dusi

**Affiliations:** 1Cardiology, Department of Medical Sciences, University of Turin, 10124 Torino, Italy; giuseppe.giannino@unito.it (G.G.); valentina.braia@unito.it (V.B.); carola.griffithbrookles@unito.it (C.G.B.); federico.giacobbe@unito.it (F.G.); fabrizio.dascenzo@unito.it (F.D.); gaetanomaria.deferrari@unito.it (G.M.D.F.); 2Division of Cardiology, Cardiovascular and Thoracic Department, ‘Città della Salute e della Scienza’ Hospital, 10126 Torino, Italy; fangelini@cittadellasalute.to.it (F.A.); andrea.saglietto@live.com (A.S.)

**Keywords:** cardiac autonomic nervous system, intrinsic cardiac nervous system, neuromodulation, ganglionated plexi ablation, cardioneuroablation

## Abstract

**Simple Summary:**

The cardiac autonomic nervous system (CANS) controls all cardiac functions in physiology and pathology. The first intracardiac level of this system, namely the intrinsic cardiac nervous system (ICNS), has long been considered a pure parasympathetic efferent station, while it is now clear that it contains all the elements of a neuronal network. In the current review, we will focus on the role of the ICNS in cardiac physiology and pathology, and on the rationale and the results of therapeutic intervention performed so far at this level of cardiac neuraxis.

**Abstract:**

The cardiac autonomic nervous system (CANS) plays a pivotal role in cardiac homeostasis as well as in cardiac pathology. The first level of cardiac autonomic control, the intrinsic cardiac nervous system (ICNS), is located within the epicardial fat pads and is physically organized in ganglionated plexi (GPs). The ICNS system does not only contain parasympathetic cardiac efferent neurons, as long believed, but also afferent neurons and local circuit neurons. Thanks to its high degree of connectivity, combined with neuronal plasticity and memory capacity, the ICNS allows for a beat-to-beat control of all cardiac functions and responses as well as integration with extracardiac and higher centers for longer-term cardiovascular reflexes. The present review provides a detailed overview of the current knowledge of the bidirectional connection between the ICNS and the most studied cardiac pathologies/conditions (myocardial infarction, heart failure, arrhythmias and heart transplant) and the potential therapeutic implications. Indeed, GP modulation with efferent activity inhibition, differently achieved, has been studied for atrial fibrillation and functional bradyarrhythmias, while GP modulation with efferent activity stimulation has been evaluated for myocardial infarction, heart failure and ventricular arrhythmias. Electrical therapy has the unique potential to allow for both kinds of ICNS modulation while preserving the anatomical integrity of the system.

## 1. Introduction

Our understanding of cardiac neuronal control, albeit still far from being complete, has dramatically evolved in the last decades. One of the turning points has been the realization that beat-to-beat reflex control of cardiac function goes far beyond a purely descendant central command due to the existence of peripheral, subcortical pathways including all the elements of neuronal arches, namely afferent neurons, interneurons (or local circuits neurons) and efferent neurons. The first hierarchy of these neuronal networks is located within the epicardial fat pads [1], and it is better known as the intrinsic cardiac nervous system (ICNS) [2,3]. The second one is located within extracardiac but intrathoracic ganglia, such as the cervical ganglia and the stellate ganglia, while the third one involves central neuronal stations (medullary and spinal cord neurons modulated by higher centers) [3]. The entire cardiac neuraxis can be imagined as a multiple-level powerhouse that continuously receives and integrates cardiac and extra-cardiac afferent inputs and uses them to generate a double (sympathetic and parasympathetic) autonomic output that finely modulates all cardiac functions. This neuronal control system shows plasticity and memory capacity and plays a pivotal role in the maintenance of cardiac homeostasis, as well as in the development and progression of cardiac pathology.

In the current review, we will focus on the role of the ICNS in cardiac physiology and pathology, and on the rationale and the results of therapeutic intervention performed so far at this level of cardiac neuraxis.

## 2. General Principles of Functioning of Cardiac Autonomic Nervous System

The two efferent branches of the cardiac autonomic nervous system (CANS), namely the sympathetic and the parasympathetic (or vagal) arm, control all mechanical and electrical cardiac functions. Additionally, sympathovagal balance also affects the onset, maintenance and decay of cardiac inflammatory responses [4]. In general, cardiac sympathetic activation is facilitatory (including a pro-inflammatory effect), whereas parasympathetic outputs are inhibitory (including an anti-inflammatory effect), with time kinetics that can be very different. Parasympathetic effects may set in very rapidly, even within one heartbeat, and decay quickly. Conversely, both the onset and the decay of sympathetic effects are more gradual, mostly requiring more than one cardiac cycle to establish. In the case of a concomitant activation, the effects are not algebraically additive, but complex interactions prevail. Such interactions may be mediated at multiple levels from the ICNS to the CNS; at the cardiac level, they include both pre-junctional and post-junctional interactions with respect to the neuro-effector junction [5].

## 3. Embryological Origin of Cardiac Innervation

Although the heart, which is the first functional organ formed during development, begins to sketch around the 19th day in the form of two small tubes formed by angioblasts derived from the splanchnic mesoderm of the cranial region, the development of cardiac innervation occurs from the fifth week, when the cardiac chambers and the arterial trunks have already formed and partly sepimented [6].

The ICNS, unlike the cardiac muscle, which is a mesoderm derivative, originates from the neuroectoderm, specifically from neural crest cells [7]. These are multipotent stem cells, whose differentiation depends strictly on their position on the axis of the embryo, such that one can distinguish along the antero-posterior axis, the cranial neural crest, the cardiac neural crest, the thoracic neural crest and the sacral one.

In particular, sympathetic neurons originate from cardiac neural crest cells, and, during neural tube folding, migrate first to the dorsal aorta, and then rostrally and caudally to form the sympathetic paravertebral chain [6].

The trafficking of these cells is mediated by the gradient of tropic factors such as ephrins and semaphorins [8]. Indeed, the Nrp1 and Sema3a knock out result in severe sympathetic ganglion dysfunction and stellate ganglia malformation [9].

Parasympathetic ganglia also originate from neural crest cells; however, unlike sympathetic ganglia, during neural tube folding, they migrate laterally to the somites and enter the forming heart [10]. The molecules involved in this trafficking are not yet fully known.

The different differentiative fate toward the sympathetic and parasympathetic nervous systems by neural crest cells depends strictly on the growth factors to which they are exposed. In fact, at first, high concentrations of bone morphogenetic protein-2 (BMP-2) [11], released by epicardial and endothelial cells, initiate the differentiative pathway toward the sympathetic nervous system, due to the increased expression of transcription factors such as paired-like homeobox 2b (PHOX2B) and achaete–scute complex homolog-1 (ASCL1), better known as **mASH1** in rodents and hASH1 in humans [12], which are necessary for the production of enzymes involved in the biosynthesis of catecholamines; low concentrations of BMP-2 limit the production of these and initiate the cell toward differentiation into the parasympathetic arm [13].

Once committed, the sympathetic fate is maintained and supported by trophic factors such as nerve growth factor (NGF) and neurotrophin-3, while the parasympathetic fate by glial cell-line-derived neurotrophic factor family of ligands (GLs), such as glial cell line-derived neurotrophic factor (GDNF), neurturin, artemin and perspehin.

Notably, the two key enzymes involved in catecholamine biosynthesis, namely dopamine β-hydroxylase (DBH) and tyrosine hydroxylase (TH), start to be expressed at the cardiac level in the early embryo stages, much earlier than post-ganglionic sympathetic neurons differentiation even begins. Targeted disruption of these genes was proved to be embryonic lethal in rats [14,15] because of heart failure, reflecting the pivotal role of catecholamine in fetal development, even in the pre-innervation stages and before chromaffin cells within the adrenal glands can synthesize them (E15.5 in mice). The major source of catecholamines in early embryo stages is the so-called intrinsic cardiac adrenergic (ICA) cells, which were first identified by Huang et al. in 1996 [16]. ICA cells are TH-immunoreactive cells that do not generate neurites. The origin, as well as the fate of these cell types after extra-cardiac production of catecholamines begins, is still unclear, but preliminary data suggest that they might be involved in the development of the cardiac conduction system and even be able to directly differentiate into pacemaker cells [17]. Some ICA cells persist in the adult ventricular myocardium, where they might act as a “backup” system in case post-ganglionic efferent sympathetic innervation is interrupted or dysfunctional [18]. For instance, the number of ICA cells was proved to increase, along with the expression of TH, DBH, and phenylethanolamine N-methyltransferase (PNMT), after denervation due to cardiac transplantation, with a potentially significant role in post-heart transplant inotropic support [19]. ICA cells might also play a cardioprotective role in the setting of ischemia/reperfusion injury, as they also express cardioprotective genes, e.g., calcitonin gene-related peptide (CGRP) [20]. In the adult mammalian heart in physiological conditions, the interplay between the mature ICNS and ICA cells is still largely unknown.

## 4. Anatomical and Functional Organization of CANS

### 4.1. Sympathetic Branch

Efferent sympathetic preganglionic neurons have their bodies in the intermediolateral column of the spinal cord (T1-T4 levels) and synapses on postganglionic neurons located in the lower cervical and upper thoracic paravertebral ganglia [21]. In humans, the lowest cervical ganglion (C8) and the highest thoracic ganglion (T1) are generally fused bilaterally to constitute the left and the right stellate ganglia, also referred to as cervicothoracic ganglia. In less than 3% of human sympathetic chains, the second thoracic ganglion (T2) is fused as well, constituting a trilobal (C8-T1-T2) stellate ganglion. The remaining cardiac sympathetic efferent innervation is provided by T2 to T4 paravertebral thoracic ganglion, albeit a minor contribution of T5 has been suggested [21].

The terminals of postganglionic sympathetic neurons synapse both directly with specialized and working cardiomyocytes and, to a much lesser extent, with adrenergic neurons in the intracardiac ganglionated plexi. Terminal sympathetic fibers arising from the right and the left paravertebral ganglia have an asymmetrical, although partially overlapping, distribution. At the supraventricular level, right-sided sympathetic control prevails on the sinus node, left-sided on the atrio-ventricular node, and atrial chambers are under bilateral sympathetic control [22]. The crucial role of right-sided sympathetic nerves in the modulation of cardiac chronotropism has been consistently demonstrated throughout the years in conscious and unconscious animal models [23,24,25], as well as in humans [26,27], and recently confirmed using sophisticated optogenetic and viral vector strategies [28]. Asymmetry is more pronounced at the ventricular level: the anterior surface of the ventricles mostly receives right-sided projections, whereas the posterior one is mostly under a left-sided sympathetic control [29,30], with a certain degree of overlapping consistently reported in canine and porcine hearts [31,32]. Accordingly, despite most sympathetic innervation and control of the left ventricle coming from the left paravertebral ganglia, a not neglectable contribution of right sympathetic nerves has been demonstrated as well [33]; the same was demonstrated in animals for the right ventricle (predominant right control but detectable left-sided effects). Whether the dominance of the coronary tree influences the right versus left distribution of sympathetic innervation is currently unknown. From an anatomical point of view, sympathetic efferent postganglionic fibers travel along coronary arteries at the subepicardial level. At the ventricular level, a double sympathetic descending nerve density gradient has been demonstrated in several animal species including pigs [34], dogs [35] and cats [36], as well as in humans [37]: from base to apex (denser in the base) and transmural (denser in the subepicardium) [37,38]. The right ventricular outflow tract (RVOT) is a remarkable regional exception, with efferent sympathetic fibers running in the canine not only in the subepicardium but also in the subendocardium [39]. Another remarkable exception is represented by the papillary muscles that, despite being endocardial structures, display a rich sympathetic innervation and were proven to actively participate in the increased inotropic responses upon sympathetic activation; notably, a canine study showed that the posterior papillary muscle is primarily under a left side control, while the anterior papillary muscle is under a double sympathetic control (left and right) [40]. Finally, the structures of the cardiac conduction system, including the sinoatrial node, the atrioventricular (AV) node and the His bundle, are extensively innervated compared to the working myocardium with both sympathetic and parasympathetic efferent fibers.

### 4.2. Parasympathetic Branch

The soma of parasympathetic cardiac pre-ganglionic neurons is based in the medulla oblongata, specifically in the dorsal vagal motor nucleus (DVMN) and in the ambiguous nucleus [21]. From here, the axons travel along the entire length of the nerve of the same name and distribute directly at the level of the intrinsic ganglionic plexi, located on the posterior surfaces of the atria and the superior aspect of the ventricles, within epicardial fat pads. Postganglionic efferent parasympathetic neurons located in individual cardiac ganglia receive preganglionic inputs from both the right and left vagal trunks.

From a functional point of view, preganglionic parasympathetic neurons of the nucleus ambiguous and of the DVMN were suggested to have a preferential, albeit not selective, control over nodal and ventricular tissue, respectively [41,42,43,44], and recent optogenetic data support this theory [45,46]. Nonetheless, several other elements including the anatomical organization of cardiac parasympathetic pre-ganglionic and post-ganglionic efferent projections, support an integrative (divergent) parasympathetic control over cardiac functions, rather than a selective (convergent) control on specific cardiac regions and/or indices [2,47]. Indeed, vagal preganglionic neurons project bilaterally to multiple ganglionated plexi (GPs), several interganglionic connections have been shown between the aggregates of intrinsic cardiac neurons [48,49,50] and, finally, postganglionic cholinergic neurons located within the various intrinsic cardiac GPs project to all cardiac regions. Yet, to further increase in complexity at the peripheral level, broad areas of preferential (but never selective) influence have been identified: nicotine-sensitive neurons of atrial GPs affect primarily, but not exclusively, atrial tissues (with a further preferential control on the two nodal structures, as will be explained later), whereas ventricular ones affect primarily, but not exclusively, ventricular tissues [51].

In humans, autoptic studies performed on subjects without known cardiovascular disease showed that most of the cholinergic fibers access the human heart through the posterior and anterior right atria, once again supporting the divergence from GPs in these areas to those located in other cardiac regions [52]. Of note, despite the antiarrhythmic properties of vagal nerve stimulation (VNS) against ventricular fibrillation first being anecdotally reported back in 1859 [53], it took almost 100 additional years to first hypothesize the existence of parasympathetic postganglionic projections to the ventricles [54,55], and even more to collect definitive evidence that vagal activation does not affect only atrial and nodal tissues, but the ventricles as well [56]. An important turning point was the demonstration that acetylcholinesterase (AChE), the enzyme responsible for hydrolyzing acetylcholine (ACh) and known for being abundant in cholinergic neurons, is sparse or absent in adrenergic or sensory neurons in the heart. Animal [56] and human [37] studies using AChE staining clearly showed that cardiac cholinergic innervation is widespread across all cardiac chambers. Finally, cholinergic muscarinic receptors were localized in both the right and the left ventricle [56,57]. Their role in modulating ventricular function may be more important in humans than in other mammals [57].

At the ventricular level, parasympathetic fiber density, like the sympathetic one, has a base-to-apex descendent gradient; yet, at the transmural level, the gradient is opposite, with a larger nerve density in the subendocardium demonstrated in pigs [58] as well as in humans [37]. Notably, the higher density of cholinergic fibers and ACh demonstrated in the atria compared to the ventricles may in part be due to the location of both pre- and postganglionic neurons in this region [56].

Figure 1 shows the overall organization of CANS, including afferent pathways.

## 5. Anatomy of the Intrinsic Cardiac Nervous System

The body and the dendrites of parasympathetic postganglionic neurons, together with pre-ganglionic endings, interneurons, afferent neurons and other types of efferent neurons, shape the intrinsic cardiac ganglia or ICNS [59]. In fact, compared to what was previously believed, the intrinsic cardiac ganglia are not only a pure efferent parasympathetic station, but contain within them all the elements of a neuronal integrative network, namely sensitive neurons, regulatory interneurons and neurons releasing all kinds of neuropeptides, including norepinephrine (NE) and others. The presence of adrenergic neurons is believed to be due in part to the legacy of resident adrenergic cells [18], which migrated during embryonic development, and in part to dedifferentiation and transdifferentiation mechanisms [60], typical of all neurons and underlying synaptic plasticity. In fact, the release of neurotransmitters is strongly influenced by the stimulus that elicits it; in this way, the more a stimulus favors an adrenergic or cholinergic output, the more the corresponding output is magnified. The presence of local NE-releasing neurons, typical of the sympathetic nervous system, casts doubt on the two-neuron model described earlier. However, given the nature of these neurons, the two models are not exclusive and, to facilitate understanding, reference can still be made to pre- and post-ganglionic neurons in the sympathetic nervous system as well.

Each cardiac ganglion is composed of a variable number of neurons ranging from 200 to 1000, which in turn are connected with adjacent ganglia by a very extensive network of interneurons, giving rise to ganglionated plexus (GP) [59]. Based on morphological characteristics, Pauza et al. divided cardiac ganglia into two types [61]: spherical (globular) and straight (plain). In the former the density of neurons is high, there are about 100–200 neurons, and they occupy very small regions at a maximum of 0.15 mm^2^; the latter consists of about 50 serially associated neurons. Both contain unipolar, bipolar and multipolar neurons.

The anatomical distribution of GPs is different across species: they are less scattered and have a lower innervation density in smaller animals (e.g., mice and rats) compared to larger ones (e.g., rabbits, sheep, pigs and dogs) [62,63]. In larger mammals, including humans, the GPs can be located in seven subregions: dorsal right atrial, ventral right atrial, ventral left atrial, left dorsal, middle dorsal, right coronary and left coronary [62,63]. GPs are mostly (75%) found on the posterior atrial surface; the remaining are located at the transition from atria to the ventricle at the level of the tricuspid and bicuspid valves, in the annular-ventricular region, on the anterior surface of the heart. There are four GPs located in the vicinity of the pulmonary veins (PVs, Figure 2): each one of them innervates the PVs and the surrounding left atrial myocardium.

In humans, a simplification of GP nomenclature has been recently proposed [64]. The term superior paraseptal ganglionated plexus (SPSGP) was proposed for the GP alternatively referred to in the literature as superior right atrial GP, right anterior GP (RAGP), or right superior GP; the SPSGP, which is located at the junction between the interatrial septum and the superior vena cava, exerts a preferential parasympathetic control over the sinoatrial node. Similarly, the term inferior paraseptal ganglionated plexus (IPSGP) has been proposed for the GP otherwise referred to as posteromedial left atrial GP (PMLGP), inferior right GP, right inferior GP, or left inferior GP. The IPSGP, which is located near the proximal coronary sinus (and around its ostium) and the postero-septal junction between the right atrium and the inferior vena cava adjacent to the posteroinferior left atrium (at the so-called pyramid space), exerts a preferential parasympathetic control over the AV node. Both the SPSGP and the IPSGP can be targeted from the right atrial, as well as from the left atrial endocardium. The ligament of Marshall (LOM) is also considered part of the ICNS. Cholinergic nerve fibers originating in the LOM innervate the surrounding left atrial structures, including the PVs, left atrial appendage and coronary sinus.

Notably, supporting the thesis that the ICNS, as a whole, acts as a distributive control center [65], some animal data suggested that following discrete ablation of one element of the ICNS, the network adapts and functional control is restored [66].

## 6. The Autonomic Neurocardiac Junction

As opposed to what was previously believed, postganglionic nerve endings can release different neurotransmitters, which can be grouped into five large families: (1) biogenic amines, including, at the cardiac level, NE and acetylcholine (ACh); (2) amino acids and cyclic nucleotides; (3) neuropeptides; (4) gases (NO or CO); and (5) lipids. The possible release of multiple neurotransmitters, acting in different, sometimes complementary ways, greatly increases the possible regulation modalities at the level of the ICNS [63]. Pre-ganglionic terminals, both sympathetic and parasympathetic, predominantly release ACh, whereas they differ markedly in post-ganglionic neurons.

### 6.1. The Sympathetic-Cardiac Junction

The main neurotransmitter released by sympathetic postganglionic neurons is NE that, together with epinephrine released by the adrenal medulla, acts on several types of cardiac adrenergic receptors (ARs) that include β1-, β2- and β3-ARs, as well as α1- and α2-ARs [67]. In the healthy human heart, β1- and β2- are the most abundant ARs expressed on cardiomyocytes, with a β1/β2 ratio of approximately 4:1, which decreases in cardiac pathology due to the prevalent loss of β1-ARs [68]. α1-ARs are more represented at the distal coronary level, where their activation has a vasoconstrictor effect. Finally, sarcolemmal α2-AR can safeguard cardiac muscle under sympathoadrenergic surge by governing Ca^2+^ handling and contractility of cardiomyocytes [69]. α2-AR signaling alters kinase–phosphatase balance, opposing β-adrenergic stimulation and leading, among other things, to suppress voltage-gated L-type Ca^2+^ channels (*I*_CaL_) through nitric oxide (NO) dependent pathways, therefore reducing Ca^2+^ overload and its potential deleterious consequences [70].

ARs are all G protein-coupled receptors (GPCRs), but with a different downstream cascade signaling: β1-ARs couple to Gs, leading to cAMP increase, while β2-and β3-ARs can act on both Gs and Gi, with the latter eliciting a reduction in cAMP; β3-ARs can also increase cGMP through eNOS activation [67,71]. Notably, several data point towards the existence of two distinct functional pools of ARs, an extra-junctional and a post-junctional one, with β1-ARs being concentrated at the post-junctional level [72,73]. These two different pools may therefore differentially mediate the cardiac effects of circulating catecholamines and neuronal sympathetic activation [74]. In physiological conditions, βAR activation leads to increased inotropism and bathmotropism [75] as a consequence of the facilitation of sarcoplasmic release of calcium by PKA activation and RyR2 modulation; increased lusitropism by phosphorylation of phospholamban and disinhibition of SERCA2a; increased chronotropism by increased conductance of HCN channels; and increased dromotropism by increased conductance of CACNA1C channels. From a purely electrophysiological point of view, βAR activation affects both depolarization (increase in excitability and conduction velocity) and repolarization (reduction in action potential duration, increase in spatial and temporal dispersion of repolarization, mostly with neuronal sympathetic activation compared to circulating catecholamines, and increase in the steepness of the slope of action potential duration restitution curve [76]).

At least three additional sympathetic co-transmitters other than NE have been identified in sympathetic post-ganglionic efferent fibers, including ATP, galanin and neuropeptide Y (NPY) [67], with co-release mainly occurring during high-frequency neuronal stimulation [77]. While ATP is rapidly metabolized, galanin and NPY are slowly diffusing molecules with a much longer half-life and duration of action compared with classical neurotransmitters [78]. Both galanin [79] and NPY receptors (Y2 type) [80] are localized on post-ganglionic vagal terminals and were implicated in the long-lasting sympathetic inhibition of vagally induced bradycardia [81]. NPY may also act directly on Y1 receptors at the ventricular level, which are coupled to both adenylyl cyclase (inhibited) and phospholipase C (activated) [82], leading, in animal models, to a steeper action potential duration restitution curve [78]. Accordingly, in Langendorff-perfused rat hearts with intact innervation, Y1 receptor activation led to a significant reduction in ventricular fibrillation threshold despite metoprolol, which was accompanied by increased amplitude and decreased duration of intracellular calcium transients [83]. Finally, NPY is also a potent vasoconstrictor [84]. ATP binds P1 and P2 receptors, which in turn are distinct P2X and P2Y [85]. ATP administration in vivo produces AV blocks via P1 (that act similarly to the M2 muscarinic receptor) [86], while via P2X and P2Y increases cellular cation concentration [87], increases inotropism and, depending on the magnitude of the stimulus, elicits three different responses: rapid depolarization, rapid depolarization followed by hyperpolarization and, finally, slow depolarization [88]. The presence of different effects depending on the receptor considered reflects the different distribution of these in the heart and is a useful feed-forward control mechanism in the presence of redundant signals.

### 6.2. The Parasympathetic-Neurocardiac Junction

The main neurotransmitter released from the parasympathetic postganglionic terminal is ACh, which exerts its action through binding to M1-M5 muscarinic metabotropic receptors [89]. Of these the one most represented at the cardiac level is M2, a Gi-coupled GPCR, whose activation reduces cAMP concentration [90]. M1 and M3, however, which are also expressed in the heart to a much lesser extent, are coupled to Gq/11 proteins and elicit the PLC pathway. Activation of M2 receptors exerts a strong negative chronotropic effect through several mechanisms, the main one being hyperpolarization through the involvement of ACh-sensitive potassium channels (IKACh) [91]. Among other neurotransmitters, the parasympathetic postganglionic terminal also releases NO, whose presynaptic action facilitates the release of ACh and inhibits that of NE, while at the postsynaptic level, it negatively modulates HCN channels [92], and VIP, which, opposite to ACh, increases cAMP concentration via VPAC1 and VPAC2, as an autoregulatory mechanism of signaling [93]. Beyond the well-characterized negative impact on chronotropism, vagal activation typically decreases several other cardiac indices including atrial contractility, atrial effective refractory period (AERP) duration (in an inhomogeneous way, therefore significantly increasing atrial refractoriness dispersion), AV conduction and ventricular contractile force [94]. While the negative chronotropic effect largely relies on a different intracellular pathway (IKACh current activation), which is the main responsibility for the different kinetics of sympathetic and parasympathetic chronotropic control, all the other effects mostly reflect cAMP inhibition, and therefore have an opposite intracellular effect compared to catecholamines, with an expected similar time onset of the effect. Of note, since the first demonstration of the existence of a functional parasympathetic innervation at the ventricular level [95], it is now well recognized that vagal activation preferentially suppress ventricular endocardial contractile function, in particular that of the papillary muscles [40]. Notably, positive inotropic effects of high dosages of ACh level were also described in isolated human [96] as well as animal cardiomyocytes [97,98]. The positive inotropic effect is likely mediated by different intracellular pathways and potentially different subtypes of muscarinic receptors. For instance, Tsuchida et al. suggested that the positive inotropic action of ACh may be mediated by the activation of the inositol trisphosphate (IP3) pathway through the stimulation of M3 receptors in the canine cardiac Purkinje fibers [99]. Notably, the existence of two pools of cholinergic muscarinic receptors, one intimately associated with vagal nerve endings (post junctional) and the other one spatially separated (extra junctional) has also been hypothesized and may account for the opposite effects of vagal nerve stimulation (negative) and exogenous acetylcholine (positive) on ventricular contractility [100].

## 7. ICNS in Pathology: Pre-Clinical Data

Cardiac pathology, independently from the etiology, typically promotes an anatomic and functional remodeling affecting the entire cardiac neuraxis, from the peripheral (ICNS) to the central level. The primum movens is typically characterized by an abnormal afferent signaling sustained by chemo and/or mechanoreceptor persistent stimulation (e.g., acute myocardial ischemia and/or ventricular dilation or abnormal contraction leading to increased wall stress) [94]. The second hit, which dramatically enhances the first one, is the reduced cardiac output, which in turn elicits the baroreceptor reflex acutely aimed at sustaining cardiac function, but chronically leading to deleterious effects. Indeed, the first and the second hit potentiate one another and concur to determine a profound autonomic imbalance, characterized by increased cardiovascular sympathetic output and decreased parasympathetic activity. The profound autonomic remodeling occurring in cardiac pathology at the higher centers, as well as in intrathoracic extracardiac centers, has been extensively described elsewhere [65]. Here, we will focus on the ICNS. Indeed, in the last years, evidence regarding the bidirectional relationship between cardiac pathology and ICNS dysfunction has accumulated, particularly in the field of myocardial ischemia and heart failure, as well as brady- and tachyarrhythmia.

### 7.1. Myocardial Infarction

The first evidence of anatomical changes in ICNS neurons associated with cardiac pathology in general, and, more specifically, with coronary artery disease, was provided by Hopkins DA et al. back in 2000 [101]. Posterior atrial ganglia were removed from eight patients with angiographic evidence of right coronary disease at the time of cardiac surgery. In 32 studied ganglia, 35% of 473 ICNS neurons displayed striking abnormalities, represented by inclusions, vacuoles and degenerative changes. Intracellular in vitro studies subsequently showed abnormal excitability (such as increased responsiveness to histamine), altered synaptic efficacy, and maladaptive changes in neurochemical phenotypes and neuromodulation in ICNS neurons derived from post-MI guinea pigs [102,103]. Yet, the functional consequences of such changes on neural signaling in vivo were still largely unknown until 2016, when Rajendran PS et al., by using advanced techniques of in vivo neuronal recording and processing, clearly showed that MI induces profound functional as well as structural remodeling of the ICNS [104]. As opposed to the attenuated afferent signaling from the infarcted zone, inputs from border and remote regions to ICNS neurons were preserved, creating a sensory border zone. Also, the transduction capacity of convergent ICNS neurons (those receiving both afferent and efferent inputs) was enhanced. Overall, functional network connectivity within the ICNS was reduced post MI.

Another very important piece in the puzzle was added in 2017 by Vaseghi et al. [105]. Using a post-MI porcine model, they showed that cardiac Ach levels, as opposed to NE levels, remain preserved in border zones and the remote viable myocardium. Nonetheless, in vivo neuronal recordings showed that parasympathetic neurons usually activated by VNS had a lower basal firing frequency, while those usually suppressed by VNS had a higher one. Also, the functional composition of convergent LCNs was different compared to control animals: LCNs receiving parasympathetic inputs only decreased, while those receiving sympathetic inputs only increased. Accordingly, VNS applied to the same post-MI model reduced ventricular arrhythmia inducibility by decreasing ventricular excitability and heterogeneity of repolarization of infarct border zones. Overall, these data suggest that, with the only exception of the scar region, the parasympathetic cardiac neuronal network remains anatomically intact after MI but undergoes a profound functional remodeling that can be restored by VNS, providing insight into its antiarrhythmic benefit. Of note, the antiarrhythmic effect of cervical VNS towards ischemia/reperfusion-related arrhythmias and post-MI-related arrhythmias, as well its potential to slow down HF progression, have been consistently documented in both anesthetized and conscious animals starting from the 70 s [106,107]. When applied during hypoxia, Ach was proved to cause a significant prolongation of action potential duration at 90% of repolarization (APD90) and a reduction in APD dispersion in Purkinje fibers, partially reversing the shortening effect of hypoxia on the repolarization. This effect was due to the inhibition of the ATP-dependent potassium current (IK-ATP), a crucial current activated during hypoxia [108].

The reason why post-MI anatomical denervation seems to mostly affect sympathetic efferent fibers while sparing the parasympathetic ones is still to be unraveled, but this different behavior might be related to the different regulating mechanisms involved in sympathetic compared to parasympathetic denervation and reinnervation, being the latter ones, as previously explained, the lesser known [105]. Accordingly, NGF plays a pivotal role in sympathetic fiber development, maintenance and sprouting, while parasympathetic nerve development is, at least in part, dependent on glial cell line-derived neurotrophic factor (GDNF) signaling [109]. Also, post-ganglionic sympathetic fibers mostly originate from neurons located farther from the heart, while cell bodies of post-ganglionic parasympathetic neurons are anatomically closer. Finally, cholinergic transdifferentiation might play a role in the preserved Ach levels observed post MI [110].

### 7.2. Atrial Fibrillation

Several studies have highlighted how the autonomic efferent innervation of the atrial myocardium is critically involved in atrial fibrillation (AF) genesis [111]. Accordingly, a high density of M2-type muscarinic receptors and β-Ars is found in atrial cardiomyocytes next to the PVs (veno-atrial junction) [112]. Several studies emphasized the synergic action of sympathetic and parasympathetic activation on AF susceptibility. An increased firing of cholinergic fibers results in vagotonic AF [113] more typical of young patients without structural heart disease, while adrenergic activation can induce sympathetic AF [114], which is more typical of older patients with an underlying heart disease.

The ICNS, beyond playing a pivotal role in AF onset and maintenance due to its parasympathetic and, to a much lesser extent, sympathetic efferent neuron content [115], was also proved to undergo a functional remodeling during high atrial rates and therefore is an integral part of the well-known concept that AF begets AF [116]. Back in 2008 [117], the group of Sunny Po, using a model of anesthetized dogs with structurally normal hearts, demonstrated that GP ablation and complete pharmacological autonomic blockade with atropine and propranolol were both able to prevent the shortening of AERP induced by short-term (6 h) rapid atrial pacing and the related AF inducibility. Using the same animal model, it was subsequently demonstrated [118] that neural firing recorded in vivo from neurons of the anterior right GP increased hour by hour during 4–6 h of rapid atrial pacing, suggesting that GPs may provide not only the trigger for AF initiation, but also contribute to the substrate for AF maintenance, particularly in the early stages of AF and in absence of structural heart disease. Accordingly, low-level VNS (50% below the bradycardia threshold), when applied during rapid atrial pacing, was able to prevent both electrical and neuronal remodeling, confirming that ICNS dysfunction induced by high atrial rates is reversible and that low-level VNS can be used to prevent AF. Based on these strong pathophysiological data, several animal studies subsequently tested the effect of pharmacological or invasive interventions on the GPs to prevent AF. Notably, even a temporary (<3 weeks) suppression of GP activity such as the one induced by the local injection of botulinum toxin (BTX) in all GPs [119], was proved to prevent the electrical remodeling (AERP shortening) induced by long-term rapid atrial pacing (6 days/week for 3 months) and to provide a long-term suppression of AF, opening interesting therapeutic possibility for non-definitive interventions on GPs including pharmacological ones.

Concerning the contribution of each single GP to overall atrial electrical properties and AF inducibility, Hou et al. [120] performed a very elegant canine study based on the assessment of atrial electrophysiological properties before and after sequential GP ablation (specifically, SLGP was ablated first, followed by ARGP and IRGP). The study mainly focused on assessing the impact of GP ablation on sinus node and AV node properties, but some data on AF inducibility were also provided. Unilateral right and left cervical VNS shortened AERP and increased AF inducibility of the atrium and pulmonary vein near the ARGP and SLGP, respectively. Acute AERP shortening and AF inducibility induced by right VNS were eliminated by ARGP ablation, whereas SLGP ablation eliminated AERP shortening but not AF inducibility induced by left VNS. Overall, the study showed that GPs act as an integrative center with distributive properties between the extrinsic CANS and the intrinsic autonomic efferent neurons, with much more complex interactions than previously anticipated. Accordingly, in 2015 Wang et al. [121] reported that the electrophysiological effects (including AF inducibility) at the atrial level of a partial right GP ablation (anterior and inferior right GP only were targeted) in the canine model were transient, and completely reverted in the long term: indeed, immediately after right GP ablation, AERP was significantly prolonged and AF inducibility abolished, and the effect persisted for 1 month. These functional effects were associated with a reduction in nerve density. Yet, anatomical, and functional parameters reverted to pre-ablation levels after 6 and 12 months. These data close the circle, confirming the fact that GPs work as a highly plastic neuronal network, with restorative properties after partial damage. Accordingly, the expression of the growth-associated protein 43 (GAP 43), a protein expressed in the growth cones of sprouting axons, was markedly upregulated after ablation, further supporting the concept that post-ganglionic reinnervation does occur after targeted GP ablation. In this case, as opposed to what was previously reported in post-MI animals, reinnervation involved both the parasympathetic and the sympathetic branches, as demonstrated by the fact that nerve density of both TH-positive and choline acetyltransferase (CHAT)-positive nerve increased. The differential mechanisms underpinning neuronal regeneration after a tissutal and/or an ICNS injury at the atrial as compared to the ventricular level are still largely unknown but are an area of ongoing research.

Finally, several pre-clinical studies showed that low-level electrical VNS, either performed at the cervical level [118,122] or through transcutaneous electrical stimulation of the auricular branch of the vagus nerve [123,124], had a favorable antiarrhythmic effect against AF caused by rapid PV and non-PV firing [125]. Specifically, low-level transcutaneous VNS, as GP ablation, reversed right atrial pacing induced atrial remodeling and inhibited AF inducibility, suggesting that it might be a potential noninvasive treatment of AF aimed at modulating GPs activity rather than destroying them. Accordingly, Yu et al. [122] provided direct evidence by neuronal recording from the ICNS, that low-level cervical VNS suppressed AF inducibility by inhibiting the neural activity of major GPs within the ICNS.

### 7.3. Ventricular Arrhythmias

As already mentioned, sympato-vagal balance is of pivotal importance for ventricular electrical stability, both in structurally normal hearts [26,126] as well as in cardiac pathology [127,128], independently from the etiology. As a general concept, at the ventricular level, sympathetic activation is pro-arrhythmic while the parasympathetic one is anti-arrhythmic, with extremely rare exceptions. The most typical are ventricular arrhythmias (Vas) that may occur in the setting of an ischemia-elicited Bezold–Jarish reflex, namely an extreme bradycardia and hypotension (due to reflex vagal activation) elicited by acute myocardial ischemia affecting the inferoposterior wall.

Notably, despite the ICNS contains all post-ganglionic parasympathetic fibers, its specific role in Vas susceptibility is still a field under active investigation. A strong push to specifically evaluate the role of ICNS in ventricular arrhythmogenesis came with the realization that GPs may be directly targeted for the treatment of AF and bradyarrhythmia. Back in 2009, the group of JA Armour, a pioneer of ICNS, reported [129] that atrial and ventricular repolarization properties were affected by both atrial as well as ventricular GP stimulation.

In 2013, He B. et al. [130] assessed the effect of GP ablation in anesthetized dogs with either structurally normal heart or previous MI. In the first group, GP ablation significantly prolonged ERP and favored electrical alternans but did not increase ERP dispersion, ADP restitution curves steepness or ERP spatial dispersion; ventricular fibrillation threshold (VFT) was not significantly affected. In the post-MI group, ventricular arrhythmias (Vas) incidence (including spontaneous VF) was higher, and ventricular fibrillation threshold VFT showed a decreased trend compared to controls. The main limit of the study was the absence of a control group with previous MI.

In 2019, another group showed [131] an increased Vas susceptibility 8 weeks after acute MI with concomitant GPs ablation (achieved by ablation of the four major GPs and the ligament of Marshall), compared to post-MI dogs without concomitant GP ablation. GP ablation was associated with longer QT and corrected QT (QTc) values, longer ventricular ERP, larger dispersion of QT, QTc and ERP, higher inducibility of Vas and lower VFT compared to the MI group. Intriguingly, GP ablation was also associated with a significant increase in the density of TH and NGF in the border zone, suggesting that ICNS may also be involved in the modulation of ventricular neuronal remodeling during and after acute MI. Specifically, these data suggest that ICNS may reduce the sympathetic nerve sprouting promoted by acute myocardial ischemia, and therefore that the pro-arrhythmic effects of GP ablation in the post-MI model may also have an anatomical base (increase in sympathetic innervation).

Another crucial piece of evidence came from the seminal work by Meyer and colleagues [132]. They showed that mechanical disruption (achieved by surgical dissection) or pharmacological blockade of all atrial GPs decreases ventricular ERP, increases the incidence and the complexity of Vas induced by standardized ventricular programmed stimulation (10% incidence in control animals versus 100% incidence in the experimental group) and decreases ventricular cAMP levels in murine structurally normal hearts. Notably, in the same paper, the authors also reported that the accidental partial atrial denervation (mostly affecting neuronal fibers and only to a lesser extent neuronal bodies in the GPs) that occurs during conventional radiofrequency (RF) ablation of AF in humans (PV isolation) was associated with both reduced parasympathetic activity and disrupted sympathetic ventricular control, supporting previous clinical evidence reporting anecdotal cases of major Vas [133,134] and even of takotsubo syndrome [135] among patients undergoing pulmonary vein isolation. Notably, in the largest available prospective study (1053 patients) specifically assessing new-onset ventricular arrhythmias (Vas) within 1 month after a first procedure of RF catheter ablation of AF [136], there were 46 patients (4.4%) who had 62 different new-onset Vas: 42 suffered premature ventricular complexes (PVC) alone and 4 had PVC coexisting with non-sustained VT. A smaller prospective clinical study including 53 patients [136] confirmed an increased incidence of PVC (they specifically searched for outflow tract (OT) PVC, the most catecholamine sensitive) among patients undergoing RF catheter ablation of AF: 6/53 patients (11%) developed OTPVC, that, in most cases (5/6), resolved within the 1-year follow-up.

In 2023, the UCLA group [137] developed a chronic porcine model of GP ablation to assess the long-term effects (at 6 weeks) on Vas susceptibility and cardiac autonomic function in resting conditions and during acute myocardial ischemia. Only two GPs were targeted: the left superior GP and the RAGP. GP ablation was associated with cardiovascular dysreflexia, reduced cardioprotective effects of VNS, increased QTc and QTc dispersion (both at rest and during sympathetic stimulation) and increased VT/VF susceptibility at rest and during acute ischemia obtained through the left anterior descending artery (LAD) ligation. To increase the translational potential of the model, GP ablation was not performed surgically but using RF endocardial ablation and a widespread used clinical protocol to map and ablate the GP (loss of local gated high-frequency stimulation response and loss of VNS response). Intriguingly, despite the acute endpoints for targeted GP ablation were all achieved, the subsequent histological study showed that only around 55% of the examined ganglia in the two GPs were effectively ablated, suggesting that partial ablation of each targeted ganglia is enough to produce functional effects. These results are different from what was demonstrated by Wang et al. [121] in the previously described canine model of right atrial GPs, where restorative properties after ablation where shown. The difference might be related to a different GP ablation extension and potentially also to differences in atrial and ventricular effects.

Finally, the only study [138] assessing the acute effects of low-intensity atrial GP stimulation on ventricular electrophysiology in normal hearts and ventricular arrhythmogenesis during acute MI showed favorable results. Specifically, a 6 h low-intensity GP stimulation protocol did not increase Vas susceptibility in normal hearts but protected against Vas during acute MI.

### 7.4. Heart Failure

Autonomic imbalance plays a pivotal role in heart failure (HF) pathophysiology, in the setting of which it is chronically maintained by the combination of an abnormal peripheral afferent signaling (both at cardiac and extracardiac levels) and a reduced cardiac output [107,139]. Available experimental data assessing ICNS abnormalities in post-MI animal models have already been extensively discussed. Pre-clinical studies specifically evaluating ICNS remodeling in non-ischemic HF models are very limited. Back in 2003, Armour’s group [140] assessed the impact of early stages of tachycardia-induced HF on the ICNS functioning in anesthetized canines. HF was induced by 2 weeks of rapid ventricular pacing, which produced a 54% reduction in the cardiac index. While afferent neurons showed a preserved capability to transduce cardiac mechano- and chemosensory inputs, and efferent neurons a preserved response to pre-ganglionic sympathetic and parasympathetic inputs, nicotine-sensitive local circuits neurons displayed a reduced capacity to influence cardiac dynamics (quantitatively and qualitatively abnormal response to nicotine in vivo matched with abnormal membrane properties in vitro). These data remain currently one of a kind. More recently, in the attempt to unravel potentially specific ICNS-related neuronal mechanisms underlying the pathophysiology of diabetic cardiomyopathy and HF, Menard et al. [141] assessed the impact of diabetes on ICNS. They showed a temporal dystrophic remodeling affecting ICNS associated with the accumulation of reactive oxygen species (ROS) as compared to controls. Notably, diabetes is well known for being associated with cardiac autonomic neuropathy, which is typically characterized by cardiac tissue denervation affecting all kinds of neuronal fibers (sympathetic, parasympathetic, and afferent fibers), but its specific impact on ICNS was still unsolved. Another group [142] specifically analyzed the impact of diabetes on pulmonary vein ganglia in a mouse model of type 1 diabetes (Akita mouse), showing a significant remodeling in both sympathetic and parasympathetic neurons, with cellular hypotrophy and a concomitant decrease in functional activities. A final sympathetic shift in the net pulmonary vein ganglia output was also reported.

Finally, from a therapeutic point of view, very recently Dyavanapalli J et al. [143] reported a novel chemogenetic approach in transgenic rats to selectively increase intracardiac cholinergic parasympathetic activity in a pressure overload-induced HF model. Chemical activation of the GP cholinergic neurons was associated with favorable left ventricular remodeling, an improved cardiac autonomic dysfunction and a 30% mortality decrease. Similar beneficial effects were also demonstrated with direct electrical stimulation of the left superior GP in a post-MI randomized canine study where acute HF was induced immediately after left anterior descending artery occlusion through 6 h of high-rate ventricular pacing [144]. A short-time (6 h) GP stimulation in these anesthetized animals was proved to inhibit cardiac sympathetic remodeling (as demonstrated by reduced levels of c-fos and NGF in the left stellate ganglion as well as GAP43 and TH proteins in cardiac peri-infarct zone) and to attenuate acute HF progression compared to control. Based on the strong pathophysiological rationale and the favorable results of the previous studies, atrial GP electrical stimulation was proposed as a potential new treatment for HF [145].

### 7.5. Heart Transplantation

Limited but intriguing data are available concerning the impact of orthotopic heart transplantation (OHT) on the ICNS. From an anatomical point of view, the ICNS of the donor’s heart is either partially (with the biatrial technique) or completely (with the bicaval technique) preserved, according to the surgical technique. On the other side, the dysfunctional ICNS of the recipient’s heart is mostly preserved only in the case of a biatrial technique. Therefore, in the setting of a biatrial transplant, the new condition of the recipient is characterized by the persistence of most of the dysfunctional native ICNS localized in the fat pads on the posterior atrial surface, and by the presence of minor parts of the normal functioning ICNS of the donor on the anterior cardiac surface. As already discussed, cardiac intrinsic neurons were shown to retain some of their functions after chronic decentralization [47]. Back in 1994 [146], the same year as the first description of the anatomy of canine ICNS, Armour’s group also reported on the capability of the ICNS to acutely modify the function of auto-transplanted canine hearts. Specifically, they showed that nicotine-sensitive adrenergic neurons that accompany the transplanted heart can still induce considerable cardiac augmentation after transplant; the response of nicotine-sensitive cholinergic neurons was also still detectable. Notably, the authors showed that after transplantation, as opposed to before, power spectral analysis of heart rate variability (HRV) and left ventricular chamber rate of pressure rise variability almost completely lost their power to predict parasympathetic and sympathetic neuronal responses. This observation is very relevant since most human studies assessing the occurrence and the amount of reinnervation after OHT have used these indexes, often providing conflicting results. In a subsequent work from the year 2000 [147], Armour’s group analyzed the role of the ICNS in maintaining cardiac function 1 year after cardiac auto transplantation in a canine model; details about the surgical technique used were scant but a modified biatrial approach was used. Nerves were identified as crossing suture lines, proving extrinsic reinnervation. Yet, from a functional point of view, inputs from extracardiac sympathetic efferent neurons were reduced and those from parasympathetic efferent preganglionic neurons were inconsistent (only half of the dogs developed bradycardia after cervical VNS). As already showed in the previous work [146], they confirmed that, even in the chronic setting, functional reinnervation did not correlate with specific power spectra derived from heart rate variability in the conscious state; ventricular beta-adrenergic receptor function was also not related to reinnervation. Transplanted angiotensin II-sensitive sympathetic efferent neurons exerted greater cardiac control than nicotine-sensitive ones. Finally, ventricular tissue had normal β-AR affinity and density but reduced catecholamine and alpha-tubulin contents. Overall, these data show that ICNS remodels after cardiac transplantation and that a direct assessment of extracardiac and ICNS behavior is required to fully understand cardiac control after transplantation.

## 8. ICNS as a Direct Therapeutic Target: Clinical Data

Growing evidence regarding the implication of ICNS in the pathophysiology of different cardiac disorders has yielded interest in the development of targeted therapeutic strategies, particularly in the field of atrial arrhythmias and symptomatic functional bradyarrhythmia. Also, preserving the anatomical integrity of the ICNS may be one of the factors to consider in the choice of the surgical technique for OHT.

### 8.1. ICNS Modulation in Atrial Fibrillation

As already discussed, the ICNS was proved to play an important role in AF initiation, maintenance and recurrences; furthermore, it was highlighted how patients displaying intense vagal responses during transcatheter PV isolation (PVI), presumably representing a surrogate for acute ganglionated plexi modulation in the PVs’ neighboring tissue, showed less arrhythmic recurrences during follow-up [148,149].

Considering these observations, it was questioned whether GPs themselves could become a target for AF ablation, either alone or in conjunction with conventional PVI. Several clinical trials, including randomized ones, tested this hypothesis and showed that GP modulation on top of PVI led to a significant decrease in AF recurrences in patients with both paroxysmal and persistent AF undergoing endocardial ablation [150,151]. Nonetheless, pooled analysis including a recent meta-analysis including only randomized control trials (RCTs) highlighted how the adjunctive effect of GPs modulation is greater in paroxysmal AF and inversely correlates with the left atrium diameter [152]. However, GP ablation alone was not superior to PVI [153]. One study assessing the 3-year freedom from atrial tachyarrhythmias showed that isolated GP ablation obtained with RF was associated with significantly worse outcomes without antiarrhythmic drug therapy when compared to circumferential PVI (34.3% versus 65.7%, *p* = 0.008) in paroxysmal AF patients [154]. Notably, the success rate of GP ablation was strongly associated with specific ablation sites, surgical techniques, localization techniques, method of access and the incorporation of additional interventions, none of which has been standardized. The long-term durability of GP ablation may be limited by the restorative functional properties of the ICNS, which, as already stressed, works as a distributive network, combined with nerve regeneration post-ablation, as suggested by animal data [121]. Yet, it must be remembered that in the setting of a virtuous circle (AF begets AF) where GPs are actively involved, even a temporary intervention may have longer-than-expected beneficial effects, as clearly demonstrated by animal models of GP transient blockade with botulinum toxin [119].

On the other hand, evidence regarding the benefit of epicardial GP ablation on top of surgical AF ablation is controversial [155,156]. In the AFACT trial, ICNS modulation through epicardial GP ablation did not show any benefit on top of thoracoscopic PVI in patients with advanced AF, but resulted in increased complications, such as procedure-limiting bleeding and pacemaker implantation [157]. These results confirm how neuromodulation can exert a prominent effect in the early phases of the arrhythmia when ICNS contribution to AF triggering is greater and atrial remodeling is not advanced.

Notably, studies evaluating endocardial extension of ablation to left atrial GPs did not show an increase in major complications. However, extending the ablation area can lead to a higher incidence of left atrium macro-reentrant tachycardias [158]. To assess the sole impact of GP ablation without the confounding effects of the associated ablation of the local atrial tissue facing the GPs on one side and of PVI on the other side, Musikantow DR et al. [159] recently assessed the feasibility and safety of epicardial ablation of the GPs with pulse field ablation (PFA) during coronary artery bypass grafting. Pre-clinical data had previously suggested that PFA, when applied at the epicardial level, is capable of selectively ablating the GPs while sparing the atrial myocardium [160,161,162]; this is the exact opposite of what happens when PFA is applied from the endocardium, in which case the GPs are spared [159]. This apparent discrepancy may be due to several factors including the distance of the GPs from the endocardial source of PF energy, the potential reduction in PF energy by the insulating effects of epicardial fat and the different susceptibility of neurons and cardiomyocytes to irreversible electroporation [161].

The preliminary, proof of principle, single-arm, NEURAL AF study [159], including a total of 23 patients and using the prolongation of the AERP as the primary efficacy outcome, demonstrated that epicardial PFA is feasible and safe, but, despite a significant effect on AERP and mean heart rate, was not associated with significant changes in HRV. Also, the post-operatory incidence of AF was similar to historical comparison cohorts (around 30%). The ability of direct epicardial delivery of PF energy to impact the GPs, which still need to be definitively confirmed, paves the way for additional research into therapeutic interventions that target neuronal tissue while sparing the myocardium and the surrounding tissues including coronary arteries and the esophagus.

Finally, transient pharmacological GP blockade with botulinum is currently under active investigation [163] for the prevention of post-operatory AF (POAF) after cardiac surgery, which represents a particular type of AF with specific mechanisms (including an inflammatory component), with favorable results reported so far in two RCT [164,165]. Notably, POAF incidence was significantly reduced at 30 days, and the beneficial effects persisted for up to 3 years after surgery.

As a whole, the positive effects of endocardial catheter-based GP ablation on top of PVI have been assessed, albeit a more uniform approach to GP identification and ablation is warranted. Yet, as already mentioned for pre-clinical studies, the recently demonstrated pre-clinical efficacy of low-level transcutaneous VNS at the auricular level in preventing AF, confirmed by preliminary clinical studies including 3 RCT in paroxysmal AF patients [166,167,168], and 1 in the setting of POAF [169], questions the entire concept of the need for GP ablation for AF prevention and treatment when an alternative noninvasive and temporary way to inhibit their GP activity is clinically available.

### 8.2. ICNS Modulation to Prevent Bradyarrhythmia

The potential role of invasive ablation of the ICNS as an innovative treatment for patients with cardioinhibitory vasovagal reflex syncope (VVS), functional AV block and functional sinus node dysfunction was first proposed in 2005 by Pachon et al. [170]. Despite being considered benign conditions, functional bradyarrhythmias can have a strong impact on quality of life (QoL) and can lead to severe physical injury. When behavioral measures and eventually drugs fail and symptoms are invalidating, implantation of a pacemaker can be considered. However, this procedure can lead to significant long-term complications, particularly in young receivers. As an attempt to mitigate parasympathetic tone and reflexes, Pachot et al. [170] developed a strategy to identify GP location and to perform a targeted RF endocardial ablation, considered enough to reach GP because atrial wall thickness ranges from 3 to 5 mm. Of note, endocardial RF ablation at the atrial level, when applied in proximity to the GPs, is expected to damage both neuronal fibers and neuronal bodies; yet, as already described, in the elegant pre-clinical model of Chung WH et al. [137], despite the acute achievement of all the procedural endpoints for targeted GP ablation, only 55% of the targeted ganglia were confirmed to be effectively ablated in the subsequent histological study. The persistence of some neuronal bodies in the targeted GP, combined with the presence of several other neuronal bodies in the remaining GPs, provides the basis for reinnervation. Areas of autonomic nerve penetration within the atrial walls were identified through spectral analysis of endocardial potentials.

The procedure—defined as cardioneuroablation (CNA)—aimed at the ablation of GPs in three different sites (the first one between the aorta and the superior vena cava, the second between the right superior pulmonary vein and the right atrium and the third at the junction of inferior vena cava, right atrium and left atrium) and resulted in promising short-term clinical outcomes in a small cohort of 21 patients with a mean age of 48 years.

Considering the encouraging results, a series of studies further assessing the outcome of CNA followed [171,172,173], with great heterogeneity both in mapping method, acute procedural success definition and extension of the ablation area. The need for a specific pre-amplifier and software for spectral analysis prompted the search for alternative methods to map GPs, such as response to high-frequency stimulation (HFS) [174], fractionated electrogram (fEGM) mapping [175] or a pure anatomical approached based on GPs localization as assessed by cardiac CT [176], eventually combined with ^123^I-metaiodobenzylguanidine (^123^I-mIBG) solid-state SPECT [177].

Independently from technical aspects, CNA showed promising results in the prevention of recurrent VVS. A recent, large meta-analysis [178] including a total of 14 studies (only 1 RCT) published between 2005 and 2021 and of 465 patients (mean age 40 years; 54% female), even though limited by an extreme heterogeneity in terms of patients’ selection, ablation strategy and mean follow-up time, showed overall freedom from syncope of 92% with higher efficacy for the biatrial and left atrial GP ablation (93% and 94%, respectively) and lower efficacy for the right atrium only approach (used only in 8% of the patients). No differences were observed according to the mapping strategy. A significant acute impact on electrophysiological parameters of the sinus and the AV node was reported, as well as a significant decrease in HRV parameters (confirming vagal denervation) that persisted up to 1 year after CNA (albeit with a trend for increase over time). In two small studies [179,180] (one using a right-side approach, the other a left side) that reported long-term (>24 months) heart rate and HRV data, recovery for each of the evaluated variables was reported. Yet, HRV recovery after 2 years was not associated with an increased risk of recurrences, once again supporting the idea that even a temporary autonomic modulation may be enough for a longer-term favorable outcome. This might be particularly the case for cardioinhibitory VVS in young patients, whose incidence typically tends to decrease over the years, suggesting a transient pathological parasympathetic overactivity that might only require a transient treatment. Concerning the safety, no major adverse events occurred, while minor adverse events were reported in 13%, mostly represented by transient inappropriate sinus tachycardia after biatrial ablations and by AF during HFS to map GPs. A large prospective study published in 2023 [172] and including 115 patients (mean age 39 ± 13 years) who received an extensive, bilateral, GP ablation, confirmed a good (83%) syncope free survival at a median follow-up of 28 months (range 12–75), with 3% of patients requiring repeated CNA and 4% requiring pacing. Notably, following CNA mean heart rate increased from 60 ± 14 to 90 ± 16 bpm, with 31 (27%) patients complaining about symptomatic inappropriate sinus tachycardia including 8 (7%) requiring chronic beta blocker and/or ivabradine and one requiring invasive sinus node modification. Other complaints included dyspnea, chronic chest pain and decreased exercise capacity which were mild and reported by 16 (14%) patients.

Finally, the latest ROMAN clinical trial [181] randomized patients with a high burden of VVS (mostly cardioinhibitory) and a positive atropine test (sinus rate increase of ≥25% at 2 min) to CNA (performed using RF ablation of the GPs from the left and right atria mapped using anatomical localization + fEGMs) versus optimal nonpharmacologic therapy (1:1 ratio). Two main GPs were targeted: the SPSGP or RAGP (from the right side) and the IPSGP or posteromedial left GP (from the left side). Overall, 48 patients (24 per arm) were enrolled (mean age 38 years, 58% female, mean of 3 syncopal episodes in the year preceding the enrollment, mostly traumatic). CNA was associated with a dramatical improvement in the 2-year syncope-free survival (92% vs. 46%, *p* < 0.001), which was the primary endpoint, as well as in QoL. As already reported by Pachon et al. in 2020 [182], the significant heart rate and HRV changes detected at 3 months after CNA persisted for up to 2 years, with 2 patients developing inappropriate sinus tachycardia treated with drugs. Pachon et al. [182] also assessed QTc and VAs at Holter ECG after CNA, showing no modification in the QT interval and the lack of a pro-arrhythmic effect at the ventricular level. Of note, the amount of resting heart rate mean increase after GP ablation, which reaches 25–30 bpm in some studies, is also not trivial and may be of concern not only because of associated patients’ symptoms, but also because of the increased risk of major cardiovascular events associated with increased resting heart rates, particularly with absolute values above 80–85 bpm [183,184].

Based on all the available clinical evidence so far, and while waiting for more efficacy as well as procedural and safety data, a very recent expert opinion document suggested that the procedure should be restricted to younger patients (<60 years) with a high burden of documented cardioinhibitory VVS, and to target the SPSGP and the IPSGB only from the right atrium, limiting a more extensive bilateral ablation to those with recurrences after the first procedure. Of note, an even more selective partial ablation only targeting the SPSGP (or right anterior GP) may mitigate the concerns about possible long-term side effects (particularly at the ventricular level) of more extensive parasympathetic denervation [185], but syncopal recurrences with a different mechanism (AV block compared to sinus arrest or block) have been reported, once again underlying the plasticity of the ICNS. From a functional point of view, the persistence of vagal activity attenuation for up to 2 years, despite the pre-clinical demonstration of earlier anatomical reinnervation occurring, may suggest that this phenomenon is insufficient to conceal the procedure’s effects. These data match with the pre-clinical model of CNA of UCLA’s group previously described and mandate extreme caution for a proper patient selection (highly symptomatic patients with a low expected VA risk for at least several years). Of course, another possible explanation is the already described intrinsically self-limiting nature of recurrent cardioinhibitory VVS syncopal episodes in young people [186].

### 8.3. ICNS Modulation in Heart Failure

As already mentioned, heart failure is associated with a significant degree of autonomic imbalance. Accordingly, several kinds of interventions of neuromodulation, mostly of an electrical nature, are currently under clinical evaluation in this setting. At present, none of them directly target the ICNS, albeit all of them, in different ways, aim at increasing cardiac vagal output and therefore ICNS activity. The most studied so far include baroreflex activation therapy (BAT), cervical vagal nerve stimulation VNS and spinal cord stimulation (SCS) [107]. BAT, albeit still lacking definite survival benefit data, is currently the only autonomic regulation therapy (ART) approved for clinical use by the FDA, while cervical VNS is still considered investigational. A possible advantage of BAT, compared with the more complex mechanism of cVNS and SCS, is its action on a well-defined autonomic afferent pathway (the baroreceptor reflex), which is functionally depressed in HF with reduced ejection fraction and a main contributor to cardiovascular autonomic imbalance. Afferent information is then integrated with other cardiovascular inputs at the central level to promote positive autonomic remodeling [94]. On the other side, cervical VNS poses some unique challenges related to the characteristics of the stimulation and the complexity of the vagus nerve composition, which were not properly addressed by the first human studies [106,107], leading to contrasting results. For an extensive discussion of the topic and of all clinical studies performed so far, the interested reader is referred elsewhere [106,107]. Finally, noninvasive, transcutaneous VNS, delivered at the tragus of the ear, where the auricular branch of the vagus nerve is located, has also been under active investigation in recent years, with promising results [187,188].

### 8.4. ICNS Preservation after OHT

The above-mentioned pre-clinical data concerning ICNS functioning after transplant overall matches with several OHT clinical studies, albeit the latter ones are often hindered but the usage of HRV as the main marker of reinnervation after OHT. Indeed, there are no reliable and reproducible ways to noninvasively assess parasympathetic innervation in vivo at the ventricular level, as opposed to sympathetic fibers distribution that can be assessed using positron emission tomography. In 1998 [189], Bernardi et al. showed that the type of surgery (biatrial in 79 subjects versus bicaval in 10 subjects) influenced parasympathetic reinnervation (assessed at the sinus node level only with power spectrum analysis), which was reported to be much more pronounced with the bicaval technique. The authors hypothesized that bicaval surgery, by cutting all sympathetic and parasympathetic extracardiac nerves, elicited stronger stimuli for regeneration in both CANS branches, while standard (biatrial) surgery, by cutting only around 50% of sympathetic fibers and a very small number of parasympathetic axons (mostly preserved), had less of such potential, particularly for parasympathetic regeneration. What they did not consider at the time was the role of the ICNS. And, although these human data matched with previous canine studies showing inconsistent VNS responses at 1 year, the impact of the recipient’s parasympathetic post-ganglionic neurons cannot be disregarded. Indeed, based on current knowledge, it can be hypnotized that the ICNS of the recipient (typically affected by advanced HF) is intrinsically dysfunctional, and therefore ICNS-promoted parasympathetic post-ganglionic reinnervation might be delayed and dysfunctional as well. Subsequent studies [190,191] showed that not only the extent but also the time course of reinnervation after OHT depended on the surgical technique. Biatrial surgery (with much more data available) was associated with significant sympathetic reinnervation (mostly assessed at the ventricular level with positron emission tomography) that began 12–18 months after OHT and increased over time; parasympathetic reinnervation began (if ever) several years after OHT, as indirectly assessed at the sinus node level using HRV. In the case of bicaval surgery (less data available) [190], sympathetic reinnervation was similar to the biatrial technique (with no significant reinnervation occurring before 6 months) but parasympathetic reinnervation occurred earlier (at 6 months) and was inversely correlated with the time of extracorporeal circulation. Finally, another study [192], based on 21 transplanted patients, showed that early parasympathetic reinnervation occurring after the bicaval technique, assessed using HRV values from 24 h Holter recordings, was not related to the reconnection of the major branches of the vagus nerve, therefore supporting the role of the ICNS of the donor.

From a clinical perspective, successful sympathetic reinnervation was associated with improved peak oxygen uptake, peak heart rate and contractile function during exercise [193], while, among patients undergoing bicaval OHT, successful parasympathetic reinnervation was accompanied by improved post-exercise heart rate recovery and quality of life [194], suggesting that preserving the ICNS of the donor may also have relevant prognostic implications. Notably, a recent large meta-analysis [195], which included 36 publications for a total of 3555 patients undergoing biatrial OHT and 3208 patients undergoing bicaval OHT, showed that early outcomes in mortality, tricuspid and mitral regurgitation and permanent pacemaker implantation, all significantly favored the bicaval technique. Long-term data were consistent, showing a significantly better survival in patients undergoing bicaval versus biatrial OHT and a significantly lower rate of late tricuspid regurgitation.

## 9. Conclusions and Future Directions

Pre-clinical and clinical data available so far clearly show that the ICNS plays a pivotal role in cardiac physiology and pathology and that its functions go far beyond those of a post-ganglionic parasympathetic station, which are still extremely important. Indeed, the ICNS is the first independent station of the cardiac ANS that allows for a local beat-to-beat, reflex control of all heart functions and responses, as well as integration and coordination with higher centers for middle- and long-term cardiovascular reflexes. From a translational point of view, more pre-clinical models, preferentially in vivo, are needed to unravel the mechanisms underlying ICNS dysfunction and regeneration following cardiac damage, as well as the effect of therapeutic interventions at this level. Advanced techniques of in vivo neuronal recording and processing and dedicated laboratories are necessary to fully unravel the complex connectivity within the ICNS, as well as its real-time changes. The differential molecular mechanisms underpinning cholinergic compared to sympathetic regeneration during and after myocardial damage are another very promising field of research. Pre-clinical models to better characterize the short- and long-term impact of ICNS function on the outcome following heart transplant are also needed. From a clinical standpoint, as it often happens in the history of medicine, the relatively easy feasibility and the good acute safety profile of GP ablation have translated into its widespread and premature clinical implementation way before long-term efficacy and safety data, as well as methods to increase accuracy and reproducibility of the procedure, were provided; nonetheless, the acute and middle-term efficacy of GP ablation, particularly for VVS, is remarkable, supporting its usage in highly symptomatic young patients without channelopathies or cardiomyopathies to avoid permanent pacing. The potential for a deleterious effect at the ventricular level (not only in the form of increased arrhythmic risk, but also in terms of overall cardiac homeostasis) is of particular concern and needs to be properly evaluated in future studies; the amount of mean resting heart rate increase after GP ablation is also of concern because of the associated increased risk of major cardiovascular events [183,184]. Notably, the integrity of the ICNS is essential for all the therapeutic interventions aimed at increasing cardiac vagal output at the ventricular level. Accordingly, the possibility of modulating ICNS activity rather than ablating the GPs should also be extensively investigated. For instance, low-level transcutaneous VNS at the auricular level has already proved to inhibit GPs and to prevent paroxysmal and post-operative AF in randomized clinical studies [166,167,168,169], questioning the concept of the need for a targeted GPs ablation, when an alternative noninvasive treatment preserving GPs integrity is available. Consistently, dedicated training programs and yoga, known for positively affecting CANS functioning, were associated with increased syncope-free survival among patients with VVS [159,196]. Finally, emerging energetic sources for invasive ablation, such as pulsed-field ablation, compared to both RF ablation and cryoablation, have the potential to create a transmural lesion while preserving neuronal structures [159] and will help to shed better light on the contribution of GP modulation to current ablative procedure outcome.

## Figures and Tables

**Figure 1 biology-13-00105-f001:**
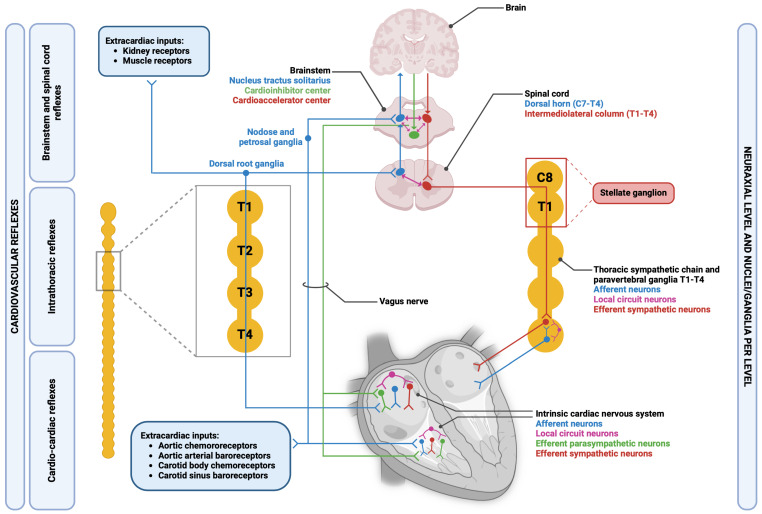
Cardiac nervous system functional and anatomical organization in humans. Blue: afferent nervous system with its ganglia and the peripheral receptors: nodose ganglia and C7-T4 dorsal root ganglia (DRG). Green: parasympathetic efferent nervous system. Red: sympathetic efferent nervous system. Purple: local circuit neurons (or interneurons). All the afferent and efferent structures except for the autonomic nuclei in the central nervous system are bilateral, although mostly represented as unilateral for simplicity. Cardiac afferent fibers traveling across the paravertebral sympathetic ganglia (usually referred to as cardiac sympathetic afferent fibers) directly reach the DRG without having synapses before. Created with Biorender.

**Figure 2 biology-13-00105-f002:**
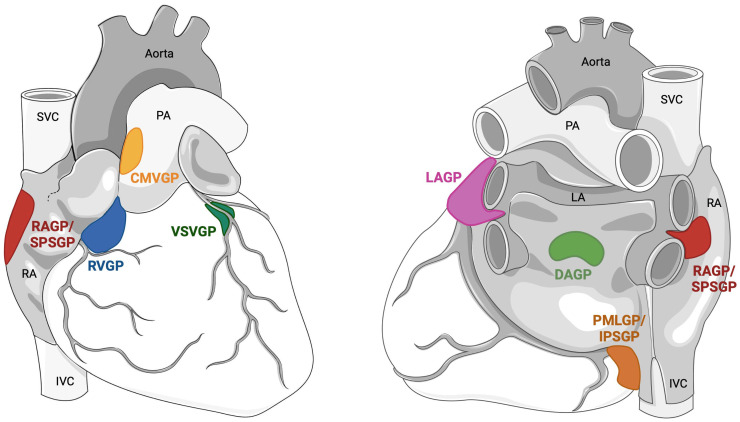
Anatomy of the intrinsic cardiac nervous system. On the left is the anterosuperior view, on the right is the posteroinferior view. The intrinsic cardiac nervous system includes all kinds of neurons, namely sensory neurons, motor neurons and local circuit neurons (or interneurons), and therefore all the basic constituents of neural circuits. These neurons are located within the ganglionated plexi (GPs) that are found in distinct areas of the heart. As extensively described in the text, GP exerts preferential, but not exclusive, influence over specific regions. For instance, the right atrial GP (RAGP, recently renominated superior paraseptatal GP or SPSGP) primarily controls sinoatrial nodal function, while the posteromedial left atrial GP (PMLGP, recently renominated inferior paraseptal GP or IPSGP) primarily controls atrioventricular nodal function. *CMVGP*: craniomedial ventricular GP, *CS*: coronary sinus, *DAGP*: dorsal atrial GP, *LA*: left atrium, *LAGP*: left atrial GP, IPSGP: inferior postero-septal GP, *PA*: pulmonary artery, *RA*: right atrium, *RVGP*: right ventricular GP, PMLGP: left atrial GP, SPSGP: superior postero-septal GP, *SVC*: superior vena cava. Created with Biorender.

## Data Availability

Not applicable.

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
