# Peer review of "The Intrinsic Cardiac Nervous System: From Pathophysiology to Therapeutic Implications"

_biology, 2024, doi:10.3390/biology13020105_

Round 1
Reviewer 1 Report
Comments and Suggestions for Authors
The authors are to be complimented on an excellent review paper in this very important space - the intrinsic cardiac nervous system. I have no significant comments.
In relation to discussions around reinnervation (after ablation) it would be interesting to see some comment about differences between neuronal cell bodies versus axons. If the neuronal cell bodies are destroyed, would axon regeneration be expected?
The authors note that GP ablation can be associated with atrial tachycardia, due to myocardial damage (140). Would be interested in their thoughts on pulsed field ablation technology that can epicardially ablate GP, but spare myocardium. https://doi.org/10.1007/s10840-023-01615-8
Some very minor typos:
Line 240: superscript 2
Line 360: 'characterized' instead of 'characterize'
Line 699: 'controversial' instead of 'controverse'
Line 837: 'not' instead of 'non'
Reviewer 2 Report
Comments and Suggestions for Authors
Very valuable, thorough, sistematic, excellent review of important, up to date knowledge in great deatils about the science of intracardiac nervous system and its promising utilization at clinical cardiac electrophysiology. Both basic scientists and clinicians will find much useful data for their field in it.
I have major and minor comments:
MAJOR
Line 197 Please cite somewhere here Kawano et al (Kawano H, Okada R & Yano K (2003). Histological study on the distribution of autonomic nerves in human heart. Heart Vessels 18, 32 – 39.)
, and please
1) emphasize a bit more that human ventricles receive parasympathetic innervation
(In the Kawano et al article six autopsied hearts without cardiovascular disease were studied by a histochemical method for acetylcholinesterase (AChE) and by an immunohistochemical method for tyrosine hydroxylase (TH).)
2) In correlation with it, please modify Figure 1 as follows:
On the schematic heart Figure (Figure 1) please draw the efferent parasympathetic neurons and the efferent sympathetic neurons in the ventricles.
This is missing.
To create the completion of these neurons, please see and study the Figure 1 from the elegant paper from Professor Coote (J. H. Coote. Myths and realities of the cardiac vagus. (2013) J Physiol 591.17 pp 4073–4085.). In that Figure 1 the parasympathetic post-ganglionic nerves are nicely shown.
For further strengthening the concept (vagal innervation of the ventricles) please cite Ulphani et al (Ulphani JS, Cain JH, Inderyas F, Gordan D, Gikas PV, Shade G, Mayor D, Arora R, Kadish AH & Goldberger JJ (2010). Quantitative analysis of parasympathetic innervation of the porcine heart . Heart Rhythm 7, 1113 – 1119.)
/Though it is porcine heart, the vagal innervation of the ventricles is shown very nicely/.
Additionally, you can consider to compose a sentence like this, taken from Prof. Casadei:
„muscarinic cholinergic modulation of ventricular function in humans may be more important than in other mammals” (Article: Casadei B. (2001). Vagal control of myocardial contractility in humans . Exp Physiol 86, 817 – 823.
Other (technical recommendation for Figure 1) : Nucleus tractus solitarius („u” was missing from tracts)
Line 160 : ……. (denser in the subepicardium) Please write in here the human data also
The human data You can find in Kawano et al ((Kawano H, Okada R & Yano K (2003). Histological study on the distribution of autonomic nerves in human heart. Heart Vessels 18, 32 – 39.)
on page 33, at results.
The data are:
„…there were more TH-positive nerves in the subepicardial than the subendocardial area of the myocardium….”
Line 202: Yes, You can also cite here Kawano et al ((Kawano H, Okada R & Yano K (2003). Histological study on the distribution of autonomic nerves in human heart. Heart Vessels 18, 32 – 39.)
/Fig. 10 in this article also shows it./
Line 302:
α2-ARs coud be also mentioned here.
You can check:
Kokoz YM, Evdokimovskii EV, Maltsev AV, Nenov MN, Nakipova OV, Averin AS, Pimenov OY, Teplov IY, Berezhnov AV, Reyes S, Alekseev AE. Sarcolemmal α2-adrenoceptors control protective cardiomyocyte-delimited sympathoadrenal response. J Mol Cell Cardiol. 2016 Nov;100:9-20. doi: 10.1016/j.yjmcc.2016.09.006. Epub 2016 Sep 19. PMID: 27659409.
Article highlights important and interesting data on:
„α2-AR in cardiomyocytes represent previously unrecognized cardiomyocyte-delimited stress-reactive targets with a potential to reduce the risk of myocardial damage.”
„α2-AR in cardiomyocytes suppress voltage-gated L-type Ca2 + channels (ICaL). NO-dependent α2-AR signaling suppresses ICaL via cGPM - PKG-dependent routes. α2-AR signaling alters kinase-phosphatase balance opposing β-adrenergic stimulation. α2-AR control [Ca2 +]in and contractility outcomes during catecholamine surge.aberrant α2-AR signaling in SHR cardiomyocytes may be linked to cardiac hypertrophy.
Figure 7 in this article deals with: Suggested main signaling pathways translating targeting of α2-AR to regulation of intracellular Ca2+ in cardiac ventricular myocytes”
Line 360 and Line 371: regarding the „ vagal activation decreases” …
I agree that this classical is correct but please also consider the possible other consequences which might also happen:
Please check and if agree, mention here some positive inotropic (in certain conditions) effects of ACh in human, cat and rat:
from authors e.g. Zipes, Braunwald etc. Suggestions below enlisted:
Du XY, Schoemaker RG, Bos E, Saxena PR. Characterization of the positive and negative inotropic effects of acetylcholine in the human myocardium. Eur J Pharmacol. 1995 Sep 15;284(1-2):119-27. doi: 10.1016/0014-2999(95)00384-w. PMID: 8549614.
Gilmour RF Jr, Zipes DP. Positive inotropic effect of acetylcholine in canine cardiac Purkinje fibers. Am J Physiol. 1985 Oct;249(4 Pt 2):H735-40. doi: 10.1152/ajpheart.1985.249.4.H735. PMID: 2996370.
Nadler E, Barnea O, Vidne B, Isakov A, Shavit G. Positive inotropic effect in the heart produced by acetylcholine. J Basic Clin Physiol Pharmacol. 1993 Jul-Sep;4(3):229-48. doi: 10.1515/jbcpp.1993.4.3.229. PMID: 8679518.
Tsuchida K, Mizukawa Y, Urushidani T, Tachibana S, Naito Y. An inotropic action caused by muscarinic receptor subtype 3 in canine cardiac purkinje fibers. ISRN Pharmacol. 2013 Oct 23;2013:207671. doi: 10.1155/2013/207671. PMID: 24260719; PMCID: PMC3821913.
Buccino RA, Sonnenblick EH, Cooper T, Braunwald E. Direct positive inotropic effect of acetylcholine on myocardium. Evidence for multiple cholinergic receptors in the heart. Circ Res. 1966 Dec;19(6):1097-108. doi: 10.1161/01.res.19.6.1097. PMID: 5928546.
Line 425: You might consider citing this:
Magyar T, Árpádffy-Lovas T, Pászti B, ……………Papp JG, Nagy N, Koncz I. Muscarinic agonists inhibit the ATP-dependent potassium current and suppress the ventricle-Purkinje action potential dispersion. Can J Physiol Pharmacol. 2021 Feb;99(2):247-253. doi: 10.1139/cjpp-2020-0408. Epub 2020 Nov 26. PMID: 33242286.
some statements might be of interest from the article e.g.: …….„When applied during hypoxia, 5 μmol/L acetylcholine caused a significant APD90 prolongation ……., partially reversing the effect of hypoxia on the repolarization. AMP returned to a normal range, while Vmax…. „
Line 655 please check and if agree you might write out for the readers the VNS as
vagal nerve stimulation as it is in Table 1 in the original article, though in the text „cervical vagosympathetic trunk stimulation” (as written on page 1776 in the original article if I understood well) is written
/ Murphy DA, Thompson GW, …..Armour JA. The heart reinnervates after transplantation. Ann Thorac Surg. 2000 Jun;69(6):1769-81. doi: 10.1016/s0003-4975(00)01240-6. PMID: 10892922./
Last major recommendation is:
As subchapter, might be written: VNS and heart failure
Some background info copied here:
„………..Vagal nerve stimulation (VNS) holds a strong basis as a potentially effective treatment modality for chronic heart failure…….Patients diagnosed with heart failure have a low vagal tone and high sympathetic activity. Vagal nerve stimulation (VNS) possesses a strong basis as a potentially effective treatment modality for chronic heart failure. Three clinical trials have been completed, i.e., the Autonomic Neural Regulation Therapy to Enhance Myocardial Function in Heart Failure (ANTHEM-HF) trial, the Neural Cardiac Therapy for Heart Failure (NECTAR-HF) trial, and the Increase in Vagal Tone in Heart Failure (INOVATE-HF) trial, and a fourth trial is ongoing, i.e., the ANTHEM-Autonomic Regulation Therapy to Enhance Myocardial Function and Reduce Progression of Heart Failure with Reduced Ejection Fraction (ANTHEM-HFrEF) trial….” from Verkerk, Doszpod….:
/Verkerk AO, Doszpod IJ, …………….Efimov IR, Wilders R, Koncz I. Acetylcholine Reduces L-Type Calcium Current without Major Changes in Repolarization of Canine and Human Purkinje and Ventricular Tissue. Biomedicines. 2022 Nov 21;10(11):2987. doi: 10.3390/biomedicines10112987. /
MINOR COMMENTS:
Line 20 epicardial ( „l” letter was missing from „epicardia” )
Line 21 contain
Line 27 studied
Line 27 arrhythmias
Line 113 cell types
Line 119 Phenylethanolamine N-methyltransferase (PNMT)
Line 167 „is” needed here instead of „in” : Correct : ….while the anterior papillary muscle is under…. / NOT : while the anterior papillary muscle in under /
Line 240: consists
Line 250: pulmonary veins (PVs)
Line 264: The ligament of Marshall (LOM) is also considered……
Line 265: ….originating in the LOM innervate…….
Line 281: inferior paraseptal Not: inferior paraseptatal
Line 292: acetylcholine (ACh) Not: achetylcholine (Ach);
Line 296: ACh Not: Ach
Line 336: decreased
Line 351: cAMP Not: AMPc
Line 355: IKACh Not: IKAch (Capital letter C is recommended)
Line 358: cAMP Not: AMPc
Line 360: characterized
Line 364: IKACh Not: IKAch (Capital letter C is recommended)
Line 402: Rajendran PS et al Not: Rajendran PS at al
Line 410: ACh Not: Ach
Line 453: maybe You can write out: atrial effective refractory period (AERP)
Line 482: Wang et al Not: Wang at al
Line 504 inhibited
Line 522 ….treatment of AF….. („of” was missing)
Line 525: He B et al
Line 529 ventricular arrhythmias (Vas)
Line 530 ventricular fibrillation threshold (VFT)
Line 602 dystrophic
Line 607 ICNS Not: INCS
Line 619 Through (not though)
Line 683 meta-analysis (Not: metanalysis) ; you might full write out randomized controlled trials (RCT)
Line 687 worse (Not: worst)
Line 693 works („work” also might be good here, but I think works is better to be used here)
Line 699 controversial
Line 729 and can lead
Line 746 please check: fractionated electrogram (fECG) - maybe (fEG) acronym is enough
or similarly that of Line 782 (fEGM)
Line 751 meta-analysis (Not: metanalysis)
Line 795 increased risk
Line 817 above-mentioned
Line 878 easy Not: easily
Line 886 amount of mean resting heart rate
Line 899 shed light on
Round 2
Reviewer 2 Report
Comments and Suggestions for Authors
I congratulate to the Dear Authors.
Very careful and nice revision.
Only 1 small typo: Line 1009: correct is:
".....to shed better light on the......"